# SARS-CoV-2 mRNA Vaccine Response in People Living with HIV According to CD4 Count and CD4/CD8 Ratio

**DOI:** 10.3390/vaccines11111664

**Published:** 2023-10-30

**Authors:** Alessandra Vergori, Alessandro Tavelli, Giulia Matusali, Anna Maria Azzini, Matteo Augello, Valentina Mazzotta, Giovanni Francesco Pellicanò, Andrea Costantini, Antonio Cascio, Andrea De Vito, Lorenzo Marconi, Elda Righi, Assunta Sartor, Carmela Pinnetti, Fabrizio Maggi, Francesca Bai, Simone Lanini, Stefania Piconi, Gabriel Levy Hara, Giulia Marchetti, Maddalena Giannella, Evelina Tacconelli, Antonella d’Arminio Monforte, Andrea Antinori, Alessandro Cozzi-Lepri

**Affiliations:** 1HIV/AIDS Unit, National Institute for Infectious Diseases Lazzaro Spallanzani IRCCS, 00149 Rome, Italy; alessandra.vergori@inmi.it (A.V.); valentina.mazzotta@inmi.it (V.M.); carmela.pinnetti@inmi.it (C.P.); simone.lanini@inmi.it (S.L.); andrea.antinori@inmi.it (A.A.); 2Icona Foundation, 20142 Milan, Italy; antonella.darminio@unimi.it; 3Laboratory of Virology, National Institute for Infectious Diseases Lazzaro Spallanzani IRCCS, 00149 Rome, Italy; giulia.matusali@inmi.it (G.M.); fabrizio.maggi@inmi.it (F.M.); 4Division of Infectious Diseases, Department of Diagnostics and Public Health, University of Verona, 37134 Verona, Italy; annamaria.azzini@univr.it (A.M.A.); elda.righi@univr.it (E.R.); evelina.tacconelli@univr.it (E.T.); 5Clinic of Infectious Diseases, ASST Santi Paolo e Carlo, Department of Health Sciences, University of Milan, 20142 Milan, Italy; matteo.augello@unimi.it (M.A.); francesca.bai@unimi.it (F.B.); giulia.marchetti@unimi.it (G.M.); 6Department of Human Pathology of the Adult and the Developmental Age “G. Barresi”, University of Messina, 98121 Messina, Italy; giovanni.pellicano@unime.it; 7Clinical Immunology Unit, Azienda Ospedaliero Universitaria delle Marche, Marche Polytechnic University, 60126 Ancona, Italy; a.costantini@staff.univpm.it; 8Department of Health Promotion Sciences, Maternal and Infant Care, Internal Medicine and Medical Specialties “G. D’Alessandro”, University of Palermo, 90127 Palermo, Italy; antonio.cascio03@unipa.it; 9Unit of Infectious Diseases, Department of Medicine, Surgery, and Pharmacy, University of Sassari, 07100 Sassari, Italy; andreadevitoaho@gmail.com; 10Infectious Diseases Unit, IRCCS Azienda Ospedaliero Universitaria di Bologna, Department of Medical and Surgical Sciences, Alma Mater Studiorum University of Bologna, 40138 Bologna, Italy; lorenzo.marconi.mn@gmail.com (L.M.); maddalena.giannella@unibo.it (M.G.); 11Microbiology Unit, Udine University Hospital, 33100 Udine, Italy; assunta.sartor@asufc.sanita.fvg.it; 12Infectious Diseases Unit, Alessandro Manzoni Hospital, ASST Lecco, 23900 Lecco, Italy; s.piconi@asst-lecco.it; 13Instituto Alberto Taquini de Investigación en Medicina Traslacional, Facultad de Medicina, Universidad de Buenos Aires, Buenos Aires C1122AAJ, Argentina; educacionmedica.api@gmail.com; 14Centre for Clinical Research, Epidemiology, Modelling and Evaluation (CREME), Institute for Global Health, UCL, London NW3 2PF, UK; a.cozzi-lepri@ucl.ac.uk

**Keywords:** HIV, PLWH, CD4 count, CD4/CD8 ratio, SARS-CoV-2 mRNA vaccine, humoral response

## Abstract

Background: Our aim was to estimate the rates of not achieving a robust/above-average humoral response to the COVID-19 mRNA vaccine in people living with HIV (PLWH) who received ≥2 doses and to investigate the role of the CD4 and CD4/CD8 ratio in predicting the humoral response. Methods: We evaluated the humoral anti-SARS-CoV-2 response 1-month after the second and third doses of COVID-19 mRNA vaccine as a proportion of not achieving a robust/above-average response using two criteria: (i) a humoral threshold identified as a correlate of protection against SARS-CoV-2 (<90% vaccine efficacy): anti-RBD < 775 BAU/mL or anti-S < 298 BAU/mL, (ii) threshold of binding antibodies equivalent to average neutralization activity from the levels of binding (nAb titer < 1:40): anti-RBD < 870 BAU/mL or anti-S < 1591 BAU/mL. PLWH were stratified according to the CD4 count and CD4/CD8 ratio at first dose. Logistic regression was used to compare the probability of not achieving robust/above-average responses. A mixed linear model was used to estimate the mean anti-RBD titer at various time points across the exposure groups. Results: a total of 1176 PLWH were included. The proportions of participants failing to achieve a robust/above-average response were significantly higher in participants with a lower CD4 and CD4/CD8 ratio, specifically, a clearer gradient was observed for the CD4 count. The CD4 count was a better predictor of the humoral response of the primary cycle than ratio. The third dose was pivotal in achieving a robust/above-average humoral response, at least for PLWH with CD4 > 200 cells/mm^3^ and a ratio > 0.6. Conclusions: A robust humoral response after a booster dose has not been reached by 50% of PLWH with CD4 < 200 cells mm^3^. In the absence of a validated correlate of protections in the Omicron era, the CD4 count remains the most solid marker to guide vaccination campaigns in PLWH.

## 1. Introduction

Since the beginning of the COVID-19 pandemic, concerns regarding the higher susceptibility of PLWH to SARS-CoV-2 infection and worse clinical outcomes have been raised, considering long-lasting HIV-driven immune dysfunction, immune-senescence and the higher prevalence and earlier incidence of age-associated comorbidities. Over time, contradictory results on the clinical outcomes of COVID-19 have emerged [1,2], but collectively, the scientific literature agrees that a lack of viro-immunological control (low CD4+T lymphocyte cell counts and unsuppressed viral load) should be considered as a risk factor for the clinical progression of COVID-19 in PLWH [3]. Hence, since 2021, PLWH have been prioritized for COVID-19 vaccination, the main measure used to reduce the severity and mortality of COVID-19. It is therefore critical to quantify the COVID-19-vaccine-induced humoral response, the peak and subsequent decline of response, particularly in the group with a higher risk of progression, in order to also further understand the potential benefits of offering additional vaccine doses.

The immunogenicity of mRNA vaccines (mRNA-1273 and BNT162b2) has been extensively studied in people living with HIV (PLWH) and has been reported to be similar to that of the general population, particularly in people with well-controlled HIV infection [4,5]. In contrast, PLWH with low CD4+T lymphocyte cell counts, detectable viremia and/or previous AIDS have weaker humoral responses [4,5,6,7,8,9,10,11,12], suggesting that they might benefit from additional vaccine doses. For this reason, the CD4 count level has been used to prioritize vaccination in PLWH.

The CD4/CD8 ratio is another well-studied marker of immune dysfunction in HIV in addition to the absolute CD4 count [13] and has also been implicated in predicting the magnitude of the T-cell immune response after natural SARS-CoV-2 infection [14]. There is also some evidence that the CD4/CD8 ratio might also have a role in predicting the vaccine response in PLWH. This was documented in a study with inactivated vaccine showing that the humoral response of PLWH with a low CD4/CD8 ratio was weaker than those with a medium or high ratio [15]. In addition, it has been shown that the inversion of the CD4/CD8 ratio (<1), despite CD4 restoration, reflects underlying immune activation and immune-senescence, deeply related to a poor vaccine response [16]. However, the role of the CD4/CD8 ratio in predicting humoral immunity remains unclear [14], and currently, the CD4/CD8 ratio is not considered in guidelines as a criterion for vaccine prioritization.

Several studies have suggested the designation of vaccine-elicited neutralizing antibody (nAbs) titers as a correlate of protection (CoP) against SARS-CoV-2 infection, and nAbs have been accepted by regulatory authorities; however, a reliable threshold has not yet been established to date [17].

Gilbert et al. attempted to derive a humoral threshold for CoPs by identifying the humoral response level that was associated with preventing 90% of symptomatic SARS-CoV-2 infections (VE90%) in the Moderna COVE phase 3 clinical trial (mRNA 1273 vaccine); this was suggested to be 775 BAU/mL (binding antibody unit/milliliter) for anti-RBD and 298 BAU/mL for anti-S on day 57. However, participants in this trial received the vaccine when Omicron was not the major circulating variant of concern (VoC) [18].

An attempt to derive the neutralization activity from the levels of binding antibodies came from a study conducted by Matusali et al. [19], which identified 870 BAU/mL for anti-RBD and 1591 BAU/mL for anti-S as the optimal cut-offs for predicting the average neutralization activity (nAbs titer ≥ 1:40) measured against the ancestral D614G strain, with 99% specificity (i.e., OCOnAbs, optimal cut-off neutralization antibody).

In this analysis, we planned to estimate the rates of not achieving a robust/above-average humoral response to COVID-19 mRNA vaccines in PLWH who received ≥2 doses, using two different endpoints indicative of the CoP/in-laboratory average response derived from binding antibodies, and to investigate the role of CD4 and the CD4/CD8 ratio in predicting a humoral response in this setting.

## 2. Materials and Methods

### 2.1. VAXICONA-ORCHESTRA Cohort

The VAXICONA-ORCHESTRA cohort is a prospective observational multicenter cohort that began in January 2021 and includes adults (≥18 years) with HIV-1 infection who have received at least two doses of COVID-19 mRNA-based vaccine. The cohort is conducted in 16 institutions, 14 from Italy, 1 from Spain and 1 from Argentina. A full list of hospitals and contributors is detailed in the acknowledgments. The cohort is included in Work Package 4 (WP4), focusing on different fragile populations, of the Horizon2020 funded ORCHESTRA project (https://orchestra-cohort.eu/; Access date: 1 September 2023), which aims to establish a pan-European cohort to rapidly advance knowledge on COVID-19. Study data were registered using the electronic case report form of the ICONA cohort (www.icona.org; Access date: 1 September 2023) and the centralized REDCap capture tool for ORCHESTRA. A common standard protocol based on the ORCHESTRA harmonization and standardization procedure for data collection was used for data exchange.

### 2.2. Study Population

PLWH of the VAXICONA-ORCHESTRA cohort who received ≥2 doses of SARS-CoV-2 mRNA vaccine and for whom ≥1 measure of anti-RBD/anti-S serology was available, together with CD4 and CD8 cell counts before 1st dose or within 1-month after 1st dose. PLWH who naturally acquired SARS-CoV2 before vaccination and during follow-up were excluded.

### 2.3. Humoral Anti-SARS-CoV-2 Response Quantification and Time Points

Vaccine-induced humoral responses were measured using a range of different assays across the centers: Elecsys^®^ Anti-SARS-CoV-2 S ECLIA assay (Roche Diagnostics, Rotkreuz, Switzerland), LIAISON^®^ SARS-CoV-2 TrimericS IgG assay and LIAISON^®^ SARS-CoV-2 S1/S2 IgG (DiaSorin, Saluggia, Italy), ARCHITECT^®^ SARS-CoV-2 IgG II Quantitative (Abbott Laboratories, Abbott Park, IL, USA) and V-PLEX SARS-CoV-2 Panel 6 Kit IgG (Meso Scale Discovery, Rockville, MD, USA).

Quantitative serology results were measured and converted where necessary to WHO binding antibody units BAU/mL, using a conversion factor provided in each manufacturer’s instructions (Appendix A).

The anti-SARS-CoV-2 serological response was evaluated at the following time points: the day of 1st vaccine dose administration (T1); the day of 2nd dose (T2, 21 days after T1 for BNT162b2 or 28 days after for mRNA-1273); 1 month after 2nd dose (T3); the day of 3rd dose (T4), 1 month from 3rd dose (T5), and 6 ± 2 months from 3rd dose (T6).

The occurrence of SARS-CoV-2 infection and the clinical course of the infection were collected at baseline and during follow-up visits using clinical information and anti-N SARS-CoV-2 data (for a subgroup of PLWH) using the six time points mentioned above and the following assays: Elecsys^®^ Anti-SARS-CoV-2 anti-N (Roche Diagnostics, Rotkreuz, Switzerland), ARCHITECT^®^ SARS-CoV-2 anti-N IgG (Abbott Laboratories, Abbott Park, IL, USA) and V-PLEX SARS-CoV-2 Panel 6 Kit IgG (Meso Scale Discovery, Rockville, MD, USA).

### 2.4. Endpoints

The Ig-G anti-SARS-CoV-2 response was evaluated at the different time points both as a continuous factors (log_2_ scale) and as a proportion of participants not achieving a robust/above-average response. A two-fold definition for a robust/above-average humoral response was used: (i) VE90%: anti-RBD level > 775 BAU/mL or anti-S > 298 BAU/mL, which corresponds to 90% vaccine efficacy according to Gilbert et al. [18] and (ii) OCOnAbs: anti-RBD level > 870 BAU/mL or anti-S > 1591 BAU/mL, corresponding to the optimal cut-off for circulating binding antibodies to identify medium neutralization activity according to Matusali et al. [19].

PLWH were stratified according to exposure groups, defined according to the CD4 count and the CD4/CD8 ratio measured when receiving their first dose: low CD4 (LCD4) if <200 cell/mm^3^, intermediate CD4 (ICD4) if >201–500 cell/mm^3^ and high CD4 (HCD4) if >500 cell/mm^3^; similarly we defined the following cut-offs, for the CD4/CD8 ratio, after calculating the tertiles of the distribution: low ratio (LR) if 0.0–0.59, intermediate ratio (IR) if 0.60–0.99 and high ratio (HR) if ≥1.0.

### 2.5. Statistical Analysis

Logistic regression was used to compare the probability of not achieving at T3 and T5 a robust/above-average anti-S/anti-RBD response above the thresholds described, according to CD4 and CD4/CD8 at the first dose as strata and as continuous measurements (per 1 standard deviation (SD) decrement). A mixed linear model with a random intercept and slope was also used to estimate the mean anti-RBD titers (using a log_2_ scale) at the six time points of the study, which were then compared across exposure groups of CD4 and the CD4/CD8 ratio. All multivariable models included the following key confounders: age, HIV-RNA ≤ 50 copies/mL at baseline, number of comorbidities and CD4 nadir. The logistic regression and linear mixed models were repeated in a sensitivity analysis which excluded values measured using the Elecsys^®^ Anti-SARS-CoV-2 ECLIA assay (which detects the total antibodies against SARS-CoV-2 S RBD and not only IgG, like the other immuno-assays). Median response anti-S/anti-RBD response was also evaluated according to the assay used for quantification. 

### 2.6. Ethics

The WP4 study of ORCHESTRA was approved for the Italian centers by the ‘Agenzia Italiana del Farmaco’ and centrally by the Ethics Committee of ‘INMI Lazzaro Spallanzani’, according to the Italian legislation for SARS-CoV-2 studies, and by each Ethics Committee of the participating centers outside Italy. The study was conducted in accordance with the Declaration of Helsinki. Informed consent was obtained from all of the subjects involved.

## 3. Results

### 3.1. Study Population

A total of 1302 PLWH met the inclusion criteria of the study; 126 PLWH had evidence of natural SARS-CoV-2 infection (10.3%) and were excluded. The remaining 1176 PLWH were distributed among the exposure groups as follows: LCD4: n = 77, ICD4: n = 298 and HCD4: n = 801 for CD4 count at baseline and LR: n = 381, IR: n = 375 and HR: n = 420 for the CD4/CD8 ratio. The general characteristics of the study population according to these groups are shown in Table 1.

Overall, the median age was 53 years (IQR 44–59), and 31% of participants had at least one comorbidity. The median time since HIV diagnosis was 11 years (IQR 5–21), the CD4 nadir was 189 cells/mm^3^ (IQR 57–348), the median of CD4 count at baseline was 606/mm^3^ (IQR 397–838) and that of the CD4/CD8 ratio was 0.8 (IQR 0.5–1.1), and 93% of participants had HIV-RNA < 50 copies/mL at baseline. The median time from the completion of the primary cycle (second dose date) to the first booster dose was 183 days (IQR 160, 212).

According to the CD4 count, LCD4 had a higher median age (*p* = 0.013) and more frequently had at least one comorbidity compared to ICD4 and HCD4 (*p* = 0.001), whereas a higher proportion of HCD4 vs. ICD4 and LCD4 were virologically suppressed at the first vaccine dose and at booster dose administration (*p* < 0.001 and *p* = 0.031, respectively). According to the CD4/CD8 ratio, no differences in age and proportions of participants with HIV-RNA < 50 cps/mL across the groups were found at the time of the booster doses.

### 3.2. Binary Humoral Response

After a median of 31 days (IQR 29–37) from the second dose, the overall proportions of PLWH not reaching the VE90% and OCOnAbs thresholds were 59.3% and 71.0%, respectively.

When investigating the association with the exposure groups, the proportions of participants failing to achieve a robust/above-average response (VE90%/OCOnAbs) 1 month after the second dose were significantly higher in participants with a lower CD4 count and CD4/CD8 ratio at baseline, following a dose–response trend. Specifically, a clearer gradient was observed for the CD4 count vs. the CD4/CD8 ratio, with the following percentages: 81.8% (not reaching the VE90% level) and 85.7% (not reaching the OCOnAbs level) for LCD4, 67.8% and 76.8% for ICD4 and 53.9% and 67.49% for HCD4 (Figure 1A, chi-square *p* < 0.0001/*p* < 0.0001); similarly, for the CD4/CD8 ratio, it was 67.2% and 76.1% for LR, 57.1% and 68.0% for IR and 54.0% and 69.0% for HR (*p* = 0.005/*p* = 0.026), respectively (Figure 1B).

After a median of 16 days (IQR 14–20) post-third dose, the proportion of participants failing to have a robust/above-average response significantly decreased by 36.5% and 43.0% overall for the VE90% and OCOnAbs thresholds, as well as in all exposure groups. To go into more detail, these proportions in the CD4 groups were 50.0% (not reaching the VE90% level) and 53.1% (not reaching the OCOnAbs level) for LCD4, 39.7% and 45.8% for ICD4 and 32.4% and 39.6% for HCD4 (Figure 1A, *p* = 0.058/*p* = 0.163); for the CD4/CD8 ratio they were 41.3% and 47.7% for LR, 34.5% and 39.6% for IR and 31.9% and 40.3% for HR (*p* = 0.223/*p* = 0.280), respectively, again with a clearer decreasing gradient associated with the CD4 count groups (Figure 1B).

In the multivariable logistic regression models, after adjusting for age, HIV-RNA ≤ 50 copies/mL at the time of the first dose vaccine, number of comorbidities and CD4 nadir, a lower CD4 count were associated with a higher risk of failing to achieve VE90% at T3. The LCD4 group, compared to HCD4, had a higher probability of not achieving VE90% (aOR = 3.36, 95%CI 1.23, 9.16); this was not shown for the ICD4 group in the adjusted model (aOR = 1.61, 95%CI 0.92, 2.81) and not shown using the OCOnAb endpoint. At T3 using the VE90% endpoint and the two exposure factors as continuous variables, per 1 SD lower in CD4 count there was a 71% higher probability of not reaching VE90% (aOR = 1.71, 95%CI 1.24, 2.36) (Table 2), while the corresponding estimate for the CD4/CD8 ratio was aOR = 1.15 (95% CI 0.90, 1.49) (Table 3), indicating a poorer prognostic value to predict the response for this marker as compared to that seen for the absolute CD4 count. The predictive value of the CD4 cell count as a marker of non-reaching a robust/above-average response with the OCOnAbs endpoint at T3 was marginally significant, with an aOR = 1.34 (95%CI 0.96, 1.85) (Table 2)

The ability to predict the T5 sub-response using VE90% and OCOnAbs endpoints, according to the baseline (T1) markers, was smaller and not statistically significant for both the CD4 cell count (Table 2) and the CD4/CD8 ratio (Table 3). In the sensitivity analysis in which Elecsys^®^ Anti-SARS-CoV-2 quantifications were excluded, the results were confirmed to be similar (Appendix A).

### 3.3. Continuous Humoral Response

We then evaluated the predicted anti-RBD responses, as continuous outcomes, by fitting a linear mixed model; we found a similar dose–response relationship across the CD4 count exposure groups at each of the examined time points, with participants in the LCD4 group showing the lowest mean anti-RDB, those in the ICD4 group showing intermediate mean anti-RDB and the highest response in the HCD4 group (Fisher test *p*-value < 0.001) (Table 4).

Of note, over T2–T4, the mean predicted vaccine-elicited humoral response remained below both the VE90% and the OCOnAbs thresholds in all of the CD4 groups. Even the HCD4 group that responded the best, on average remained below the robust/above-average response up to T5. The fact that the third dose was crucial for achieving a robust/above-average response, at least for the ICD4 and HCD4 groups, was confirmed after adjusting for age, HIV-RNA ≤ 50 copies/mL at T0, nadir of CD4 count and numbers of comorbidities (Figure 2A).

Similarly, the CD4/CD8 ratio groups also showed an association with the humoral response at all of the evaluated time points, with lower responses in LR compared to both IR and HR at each time point (*p* = 0.009) (Table 4). By fitting a multivariable mixed linear model, all of the CD4/CD8 ratio groups remained, on average, below both of the chosen thresholds until the administration of the third dose (Figure 2B).

Again, analyses were repeated after excluding the Elecsys^®^ Anti-SARS-CoV-2 measurements, confirming the main results (Appendix A). Appendix A reports the unadjusted medians and IQR at different time points after stratifying for the assay used, while the median values of the anti-S/anti-RBD response have only been further stratified according to the CD4 and CD4/CD8 strata, only for ARCHITECT^®^ anti-SARS-CoV-2 IgG II and LIAISON^®^ SARS-CoV-2 IgG (Appendix A).

## 4. Discussion

Our data show that after three doses of an mRNA vaccine, 90%VE was not achieved by 50% of PLWH, classified as LCD4, versus 40% in ICD4 and 32% in HCD4. A similar association was observed when using the alternative thresholds based on average neutralization. When analyzing the risk of not achieving robust/above-average responses and the mean level of responses after the primary cycle (but not after the third dose), we found a strong association between the baseline CD4 count and humoral response to the vaccine, after controlling for key confounders.

These findings confirm our hypothesis that, given the immune dysregulation that characterizes chronic HIV infection, despite effective ART, PLWH with a low CD4 count remain at risk of lower immune responses to SARS-CoV-2 vaccination, at least in the short term, with potentially negative implications for clinical outcomes. Mechanistically, it is known that the induction of strong antibody responses is dependent on both CD4+ and CD8+ T-cell responses, and there are reports of immunocompromised individuals, including PLWH, who did not mount antibody responses to vaccination against SARS-CoV-2 and subsequently became infected [5,8,20,21,22,23,24,25].

However, our analysis also shows that in the short term after the primary vaccination cycle, the CD4 count was a stronger predictor of the humoral vaccine response than the CD4/CD8 ratio, thus confirming that the CD4/CD8 ratio has a secondary role. The association was independent of CD4 nadir [26]; however, the role of CD4 nadir has not been investigated in this study and instead was merely treated as a confounder of the main associations of interest. It is well known that HIV infection is characterized by a profound disruption of the cellular and humoral components of the adaptive immune system [27] and, PLWH with low CD4 counts were found to have weaker humoral and T-cell responses to mRNA vaccines [5], suggesting that they may benefit from additional vaccine doses. In this respect, we previously showed that a third dose of an mRNA vaccine following the primary cycle can strongly boost humoral but not T-cell responses in PLWH with advanced disease at the time of HIV diagnosis [28].

In addition, our analysis shows that, regardless of the threshold used, the third mRNA dose was pivotal in achieving a robust/above-average humoral response, at least in participants with a baseline CD4 count > 200 cells/mm^3^, as shown in previous works with a smaller sample size [29]. It should also be noted that, despite the SARS-CoV-2 humoral response elicited by the vaccine, may not reach the targets for a valid correlate of protection against infection and severe disease, is likely to be prevented by strong cellular immunity; this is also the case for the newly circulating variants of concern [30,31].

There are several factors that are potential common causes of the studied HIV markers and the levels of vaccine humoral responses, but based on our assumptions, all of the known measured potential confounders (age, baseline HIV-RNA, number of comorbidities and nadir CD4 count) were controlled for during the analysis stage. We did not control for time since HIV diagnosis because it is known to be an inaccurate estimate of the duration of HIV infection [32].

Our analysis has a number of limitations. First, this is an observational setting and unmeasured confounding bias cannot be ruled out. Second, this is a large cohort using different serology assays, across centers, with potential problems related to a lack of standardization and interpretability. However, a strong correlation between anti-S and anti-RBD was observed for values < 4000 BAU/mL [19], which are below the thresholds used in our study, and the results of the sensitivity analyses conducted after excluding measurements quantifying the total Ab (instead of IgG only) were consistent with those of the main analyses. Third, the study period covered the waves of the COVID-19 pandemic mainly sustained by Alpha/Delta circulating VoCs; therefore, it is possible that the proportion of PLWH not achieving a robust/above-average response, for both thresholds, would have been even higher for Omicron. Indeed, the vaccine efficacy threshold was defined in the COVE clinical trial, during the circulation of ancestral strains of SARS-CoV-2. During the Omicron era, the binding antibody threshold of vaccine efficacy should supposedly be substantially higher. Fourth, this study only included PLWH, without using a control group of HIV-negative people vaccinated in the same period and with assessments of the SARS-CoV-2 humoral response with the same timelines; we, therefore, cannot estimate the effect of HIV infection per se, but can only compare the response over time according to the HIV immunological markers also related to the vaccine response. Further, cross-reactivity with other common coronavirus was not measured in this study. Nevertheless, because of the established high positive predictive values of the assays used and the low circulations of other coronavirus during the pandemic, cross-reactivity was unlikely to have introduced a large bias in this analysis. Lastly, the use of two different sets of thresholds may be confusing for the reader. It needs to be made clear that the context in which these threshold values were derived and their significance are quite different. The first set of thresholds refers to vaccine efficacy in vivo. In contrast, the second set was derived in a laboratory and refers to the average response seen in the general population. Although a consistent relationship between neutralizing antibodies and clinical protection has been shown, different immunological factors contribute to vaccine efficacy, including innate and adaptive cellular responses, mucosal immunity and non-neutralizing antibody functions. Moreover, an exact protective neutralization titer threshold at which an individual is likely to be protected from getting infected with COVID-19 does not exist [33]. Indeed, the identification of such a threshold has proven difficult for a variety of reasons, such as the diversity of the assays used and the emergence of new and more immune-evasive viral variants, as well as the diversity in the vaccine platforms used [33]. Our analysis is an attempt to provide further data comparing the predictive value of laboratory- and clinically derived thresholds. Of note, the CD4 count was slighter more predictive of the VE than of the laboratory-derived response.

The major strengths of our study are the analysis of a multinational prospective cohort adopting the same protocol with the same time points, and the large sample size of PLWH with CD4 count < 200 cells/mm^3^, allowing us to better investigate the predictive role of this marker than what has previously been achieved [34].

## 5. Conclusions

In conclusion, under the assumption that the titer thresholds used are reasonable CoPs in the Omicron era as well, we found that the third mRNA vaccine dose is key in establishing protection against SARS-CoV-2 infection in most PLWH. Furthermore, the CD4 count measured before the primary vaccination cycle strongly predicts the early humoral response in PLWH better than a concomitant measure of the CD4/CD8 ratio; therefore, the absolute CD4 count should still be the HIV marker of choice to guide vaccination and booster campaigns for the more immunosuppressed PLWH. However, similar analyses need to be repeated and the detected associations should be validated in the current setting of the circulating Omicron variants.

## Figures and Tables

**Figure 1 vaccines-11-01664-f001:**
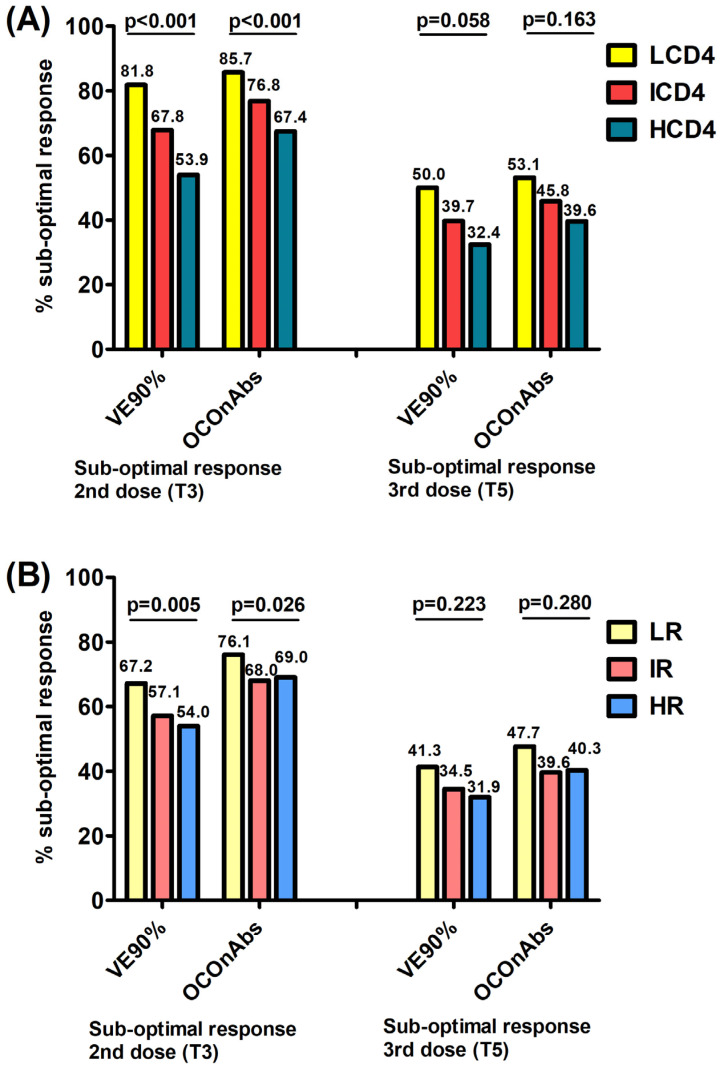
Proportions of those failing to achieve robust/above-average responses at T3 and T5, using 2 endpoints VE90% (vaccine efficacy of 90% in preventing symptomatic infections) and OCOnAbs (optimal cut-off for binding Ab calculated for a nAb titer ≥ 1:40) according to the CD4 count (**A**) and to the CD4/CD8 ratio (**B**) at the time of first vaccination.

**Figure 2 vaccines-11-01664-f002:**
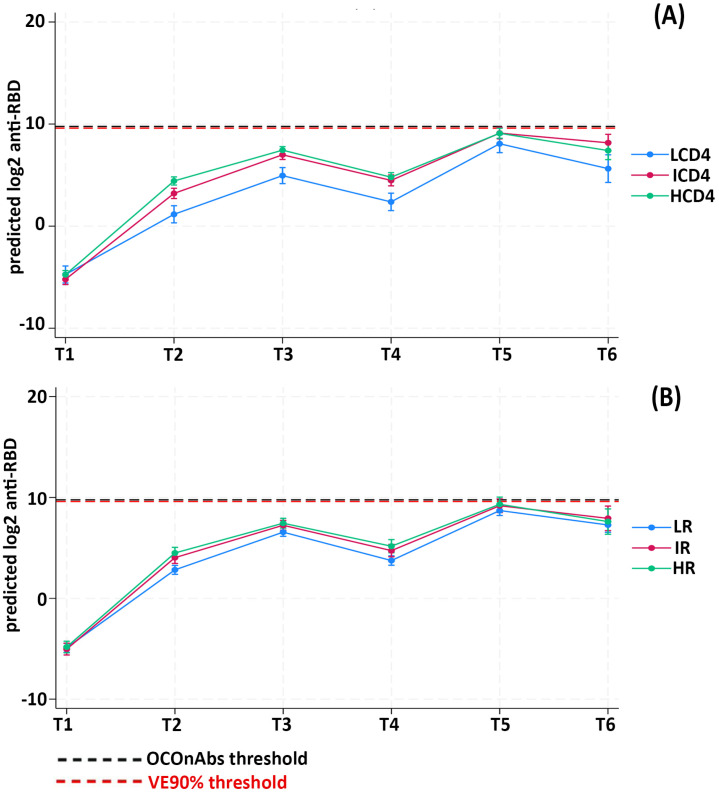
Linear mixed model prediction of log2 transformed anti-RBD titers according to CD4 strata (**A**) and CD4/CD8 strata (**B**).

**Table 1 vaccines-11-01664-t001:** General characteristics of the target population according to CD4 count (left panel) and CD4/CD8 ratio (right panel) at the time of first vaccination.

	CD4 Count at the Time of First Vaccination				CD4/CD8 Ratio at the Time of First Vaccination				
Characteristics	LCD4 N = 77	ICD4 N = 298	HCD4 N = 801	*p* *	LR N = 381	IR N = 375	HR N = 420	*p* *	Total N = 1176
**Female, n (%)**	14 (18.2)	55 (18.5)	175 (21.8)	0.397	66 (17.3)	71 (18.9)	107 (25.5)	0.010	244 (20.7)
**Age, years, median (IQR)**	56 (48, 59)	54 (45, 61)	52 (44, 58)	0.013	53 (44, 60)	53 (45, 59)	52 (44, 59)	0.485	53 (44, 59)
**Caucasian, n (%)**	57 (74.0)	240 (80.5)	747 (93.3)	<0.001	307 (80.6)	348 (92.8)	389 (92.6)	<0.001	1044 (88.8)
**≥1 comorbidity, n (%)**	34 (44.2)	108 (36.2)	224 (28.0)	0.001	129 (33.9)	130 (34.7)	107 (25.5)	0.008	366 (31.1)
**N. of comorbidities ^&^, median (IQR)**	1 (1, 2)	1 (1, 2)	1 (1, 2)	0.964	1 (1, 2)	1 (1, 2)	1 (1, 2)	0.376	1 (1, 2)
**AIDS, n (%)**	30 (39.0)	130 (43.8)	208 (26.1)	<0.001	164 (43.0)	110 (29.5)	94 (22.5)	<0.001	368 (31.4)
**Time from HIV diagnosis, years, median (IQR)**	11 (1, 24)	9 (4, 21)	11 (6, 21)	0.005	7 (3, 21)	12 (7, 21)	11 (7, 21)	<0.001	11 (5, 21)
**Time from AIDS diagnosis, years, median (IQR)**	4 (2, 9)	5 (2, 9)	8 (5, 12)	0.019	5 (2, 7)	7 (4, 10)	11 (7, 19)	<0.001	6 (3, 10)
**Nadir CD4 count, cells/mm^3^, median (IQR)**	39 (12, 72)	87 (34, 177)	272 (115, 412)	<0.001	68 (28, 166)	217 (92, 320)	312 (155, 464)	<0.001	189 (57, 348)
**CD4 count at first vaccination, cells/mm^3^, median (IQR)**	116 (61, 154)	357 (275, 436)	746 (587, 944)	<0.001	349 (206, 525)	622 (479, 790)	807 (628, 1015)	<0.001	606 (397, 838)
**CD4 count at first booster dose, cells/mm^3^, median (IQR)**	160 (109, 210)	378 (296, 443)	749 (596, 944)	<0.001	385 (228, 509)	678 (477, 837)	798 (600, 1034)	<0.001	597 (391, 832)
**CD4/CD8 ratio at first vaccination, cells/mm^3^, median (IQR)**	0.2 (0.1, 0.2)	0.5 (0.3, 0.8)	1.0 (0.7, 1.3)	<0.001	0.4 (0.2, 0.5)	0.8 (0.7, 0.9)	1.3 (1.1, 1.6)	<0.001	0.8 (0.5, 1.2)
**CD4/CD8 ratio at first booster dose, cells/mm^3^, median (IQR)**	0.2 (0.1, 0.3)	0.5 (0.4, 0.7)	0.9 (0.7, 1.3)	<0.001	0.4 (0.3, 0.5)	0.8 (0.7, 0.9)	0.0 (0.0, 1.3)	<0.001	0.0 (0.0, 1.5)
**HIV-RNA at first vaccination ≤ 50 cps/mL, n (%)**	50 (64.9)	268 (90.2)	778 (97.3)	<0.001	328 (86.1)	361 (96.5)	407 (97.1)	<0.001	1096 (93.4)
**HIV-RNA at first booster dose ≤ 50 cps/mL, n (%)**	33 (84.6)	111 (94.1)	282 (95.3)	0.031	147 (93.0)	141 (93.4)	138 (95.8)	0.542	426 (94.0)
**Having Antiretroviral Therapy**	73 (94.8)	293 (98.3)	788 (98.5)	0.064	373 (97.9)	368 (98.4)	413 (98.3)	0.853	1154 (98.2)
**Vaccination times (days), medians (IQR)**									
**From second dose to response**	30 (28.0, 31.0)	31 (30.0, 35.0)	31 (29.0, 39.0)	0.009	31 (30.0, 35.0)	31 (29.0, 37.0)	31 (29.0, 39.0)	0.805	31 (29.0, 37.0)
**From second dose to third dose**	158 (154.0, 166.0)	177 (152.0, 209.0)	189 (168.0, 216.0)	<0.001	167 (154.0, 201.5)	187 (166.0, 217.0)	190 (171.0, 213.0)	<0.001	183 (160.0, 212.0)
**From third dose to response**	17 (15.0, 20.5)	16 (14.0, 20.0)	16 (14.0, 19.0)	0.094	16 (14.0, 20.0)	16 (14.0, 20.0)	16 (14.0, 19.0)	0.339	16 (14.0, 20.0)

^&^ Comorbidities considered: chronic kidney disease, chronic obstructive pulmonary disease, myocardial infarction, congestive heart failure, peripheral vascular disease, cerebrovascular disease, dementia, chronic cognitive deficit, connective tissue disease, peptic ulcer disease, chronic liver disease and end-stage liver disease, diabetes mellitus and diabetes with organ damage, hemiplegia, solid cancers with or without metastasis, hematological cancer; * chi-square or Kruskal–Wallis test, as appropriate.

**Table 2 vaccines-11-01664-t002:** Association with CD4 count at the time of first vaccination. Odds ratio and adjusted odds ratio, by fitting a logistic regression model, of the probability of failing to achieve a robust/above-average anti-S/RBD response post-vaccination, with VE90% (Panel A) and OCOnAbs (Panel B) endpoints.

Panel A (VE90% Endpoint)	Unadjusted Odds Ratio (95% CI)	*p*-Value	Adjusted * Odds Ratio (95% CI)	*p*-Value	^&^ Type III *p*-Value
**CD4 count at T1**	**Failed to achieve VE90% after primary cycle (T3)**				
**HCD4**	1		1		0.029
**ICD4**	1.80 (1.36, 2.38)	<0.001	1.61 (0.92, 2.81)	0.097	
**LCD4**	3.84 (2.12, 6.97)	<0.001	3.36 (1.23, 9.16)	0.018	
**Per 1 SD lower (log_2_ scale)**	1.65 (1.42, 1.91)	<0.001	1.71 (1.24, 2.36)	0.001	
**CD4 count at T1**	**Failed to achieve VE90% after first booster dose (T5)**				
**HCD4**	1		1		0.937
**ICD4**	1.37 (0.89, 2.13)	0.157	0.91 (0.41, 2.03)	0.814	
**LCD4**	2.00 (1.08, 3.72)	0.028	0.81 (0.24, 2.75)	0.733	
**Per 1 SD lower (log_2_ scale)**	1.42 (1.16, 1.74)	<0.001	1.19 (0.78, 1.83)	0.421	
**Panel B (OCOnAbs Endpoint)**	**Unadjusted OR** **(95% CI)**	** *p* ** **-Value**	**Adjusted * OR (95% CI)**	** *p* ** **-Value**	** ^&^ ** ** Type III *p*-Value**
**CD4 count at T1**	**Failed to achieve OCOnAbs after primary cycle (T3)**				
**HCD4**	1		1		0.367
**ICD4**	1.60 (1.18, 2.18)	0.003	1.53 (0.84, 2.78)	0.162	
**LCD4**	2.90 (1.51, 5.58)	0.001	1.30 (0.47, 3.54)	0.612	
**Per 1 SD lower (log_2_ scale)**	1.46 (1.25, 1.72)	<0.001	1.34 (0.96, 1.85)	0.082	
**CD4 count at T1**	**Failed to achieve OCOnAbs after first booster dose (T5)**				
**HCD4**	1		1		0.925
**ICD4**	1.29 (0.84, 1.98)	0.244	0.87 (0.40, 1.89)	0.730	
**LCD4**	1.72 (0.93, 3.19)	0.083	0.83 (0.26, 2.67)	0.759	
**Per 1 SD lower (log_2_ scale)**	1.35 (1.11, 1.65)	0.003	1.19 (0.79, 1.80)	0.409	

* Adjusted for age, VL ≤ 50 cps/mL at 1st vaccination, no. of comorbidities and nadir CD4 count; ^&^ from the adjusted model; abbreviations: HCD4, high CD4 count (>500/mm^3^); ICD4, intermediate CD4 count (200–500/mm^3^); LCD4, low CD4 count (0–200/mm^3^); OR, odds ratio.

**Table 3 vaccines-11-01664-t003:** Association with CD4/CD8 ratio at time of first vaccination. OR and AOR, by fitting a logistic regression model, of the probability of failing to achieve a robust/above-average anti-S/RBD response post-vaccination, with VE90% (Panel A) and OCOnAbs (Panel B) endpoints.

Panel A (VE 90% Endpoint)	Unadjusted OR (95% CI)	*p*-Value	Adjusted * OR (95% CI)	*p*-Value	^&^ Type III *p*-Value
**CD4/CD8 ratio at T1**	**Failed to achieve VE90% after primary cycle (T3)**				
**HR**	1		1		0.311
**IR**	1.13 (0.85, 1.50)	0.393	0.76 (0.42, 1.36)	0.356	
**LR**	1.74 (1.31, 2.32)	<0.001	1.17 (0.63, 2.17)	0.612	
**Per 1 SD lower (log** ** _2_ ** ** scale)**	1.28 (1.14, 1.44)	<0.001	1.15 (0.90, 1.49)	0.267	
**CD4/CD8 ratio** **at T1**	**Failed to achieve VE90% after first booster dose (T5)**				
**HR**	1		1		0.474
**IR**	1.12 (0.67, 1.89)	0.659	0.58 (0.23, 1.46)	0.247	
**LR**	1.50 (0.92, 2.45)	0.106	0.82 (0.33, 2.05)	0.677	
**Per 1 SD lower (log** ** _2_ ** ** scale)**	1.31 (1.09, 1.57)	0.004	0.98 (0.68, 1.41)	0.921	
**PANEL B** **(OCOnAbs Endpoint)**	**Unadjusted OR (95% CI)**	** *p* ** **-Value**	**Adjusted OR (95% CI)**	** *p* ** **-Value**	** ^&^ ** ** Type III *p*-Value**
**CD4/CD8 ratio** **at T1**	**Failed to achieve OCOnAbs after primary cycle (T3)**				
**HR**	1		1		0.912
**IR**	0.95 (0.71, 1.29)	0.751	0.91 (0.50, 1.64)	0.745	
**LR**	1.43 (1.04, 1.95)	0.026	1.02 (0.54, 1.93)	0.949	
**Per 1 SD lower (log** ** _2_ ** ** scale)**	1.18 (1.05, 1.34)	0.007	0.99 (0.76, 1.30)	0.964	
**CD4/CD8 ratio** **T1**	**Failed to achieve OCOnAbs after first booster dose (T5)**				
**HR**	1		1		0.396
**IR**	0.97 (0.59, 1.60)	0.900	0.65 (0.27, 1.56)	0.332	
**LR**	1.35 (0.84, 2.16)	0.216	1.06 (0.44, 2.55)	0.898	
**Per 1 SD lower (log** ** _2_ ** ** scale)**	1.27 (1.06, 1.52)	0.009	1.09 (0.77, 1.54)	0.635	

* Adjusted for age, VL ≤ 50 cps/mL at 1st vaccination, no. of comorbidities and nadir CD4 count; ^&^ from the adjusted model. Abbreviations: HR, high CD4/CD8 ratio (>1); IR, intermediate CD4/CD8 ratio (0.60–0.99); LR, low CD4/CD8 ratio (0–0.59); OR, odds ratio.

**Table 4 vaccines-11-01664-t004:** Predicted anti-RBD adjusted means (log_2_ scale) from fitting a linear mixed model.

Anti-RBD (log_2_) Adjusted * Means	T1 95% CI	T2 95% CI	T3 95% CI	T4 95% CI	T5 95% CI	T6 95% CI	*p*-Value ^§^
** *CD4 at T1* **							<0.001
**LCD4**	−4.7 (−5.5, −3.9)	1.1 (0.3, 1.9)	5.0 (4.2, 5.8)	2.5 (1.6, 3.3)	8.0 (7.1, 8.9)	5.6 (4.2, 7.0)	
**ICD4**	−5.2 (−5.7, −4.7)	3.2 (2.7, 3.7)	9.1 (8.5, 9.6)	7.0 (6.5, 7.5)	8.1 (7.3, 9.0)	4.4 (3.9, 5.0)	
**HCD4**	−4.8 (−5.2, −4.4)	4.4 (4.0, 4.8)	9.1 (8.6, 9.6)	7.4 (7.1, 7.8)	7.7 (6.8, 8.6)	4.8 (4.3, 5.2)	
** *CD4/CD8 ratio at T1* **							0.009
**LR**	−4.9 (−5.3, −4.4)	2.8 (2.4, 3.3)	6.6 (6.2, 7.0)	3.8 (3.3, 4.2)	8.7 (8.2, 9.2)	7.3 (6.5, 8.0)	
**IR**	−5.0 (−5.6, −4.4)	4.0 (3.4, 4.5)	9.2 (8.5, 9.8)	7.2 (6.7, 7.7)	8.0 (6.8, 9.3)	4.7 (4.1, 5.3)	
**HR**	−4.8 (−5.4, −4.3)	4.4 (3.9, 5.0)	9.3 (8.6, 10.0)	7.4 (6.9, 7.9)	7.7 (6.4, 9.0)	5.1 (4.5, 5.8)	

§ F-test *p*-value for type 3 interaction; * Model adjusted for age, VL ≤ 50 cps/mL at 1st vaccination, no. of comorbidities and nadir CD4 count. time points: T1: before vaccination; T2: at the time of 2nd dose; T3: 1 month after 2nd dose; T4: at the time of the 3rd dose; T5: 1 month after the 3rd dose; T6: 6 months after the 3rd dose.

## Data Availability

The datasets generated during the current study are not publicly available because they contain sensitive data to be treated under data protection laws and regulations. Appropriate forms of data sharing can be arranged after a reasonable request to the corresponding author.

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
