# Peer review of "SARS-CoV-2 mRNA Vaccine Response in People Living with HIV According to CD4 Count and CD4/CD8 Ratio"

_vaccines, 2023, doi:10.3390/vaccines11111664_

Round 1
Reviewer 1 Report
Comments and Suggestions for Authors
The manuscript by Vergori et al. reports on humoral immune responses to SARS-CoV-2 mRNA vaccines in a large multicenter cohort of almost 1200 people living with HIV (PLWH). Demographic, immunological and virological information on the study participants was compiled and antibody levels against SARS-CoV-2 spike (S) and receptor binding domain (RBD) were measured with commercial quantitative ELISA kits. Study subjects were grouped on the basis of absolute CD4+ T lymphocyte counts and their CD4+/CD8+ T cell ratios. Consistent with other studies, the authors report that lower CD4+ T cell counts are associated with a greater likelihood of not reaching what are considered to be protective levels of antibodies against SARS-CoV-2 S protein. A similar, but statistically weaker association was found with lower CD4+/CD8+ T cell ratios. The statistical analyses appear to have been done correctly and the large size of the cohort inspires confidence in the findings. A number of limitations of the study are brought up and considered by the authors in their discussion. While the reported findings are not completely novel, the power and rigor of the study are strong. I have a number of minor comments that could improve the manuscript if addressed.
1. The groupings based on CD4+ T cell counts are well matched for virological parameters, but the groups with lower counts tended to be older, infected with HIV for longer and more likely to suffer at least 1 co-morbidity. The potential impact on the authors’ findings and analyses should be discussed.
2. If there was no significant association between nadir CD4+ T cell counts and vaccine responses, the authors should discuss what this means in terms of immune recovery for PLWH.
3. There was a weaker association between CD4+/CD8+ T cell ratios and vaccine responses than with absolute CD4+ T cell counts. The authors should discuss this and clarify how the 2 measures (CD4+/CD8+ T cell ratios and absolute CD4+ T cell counts) can each have relevant, but different meanings, especially when CD8+ T cell counts are elevated well above normal.
4. Subjects were classified as having been infected with SARS-CoV-2 on the basis of anti-nucleocapsid protein antibodies. As there is considerable cross-reactivity with the nucleocapsids of common coronaviruses (especially OC43), did the authors test for selective reactivity with SARS-CoV-2 nucleocapsid?
5. On line 78, it states a lower level of anti-S antibodies than anti-RBD antibodies was associated with protection (298 versus 775 BAU). This doesn’t make sense to me if the neutralizing antibodies are against the RBD. Please verify the accuracy of the numbers. On lines 82 and 83, a different study is cited with an opposite and more rational finding of a higher level of anti-S than anti-RBD antibodies being required for a protective effect (1591 versus 870 BAU).
6. A number of studies have suggested that the Moderna SARS-CoV-2 mRNA vaccine formulation may be slightly more immunogenic than the Pfizer SARS-CoV-2 mRNA vaccine formulation. Can the authors address this possibility with the data collected?
Comments on the Quality of English Language
Minor changes required for English language quality.
Author Response
Please see the attachment
The manuscript by Vergori et al. reports on humoral immune responses to SARS-CoV-2 mRNA vaccines in a large multicentre cohort of almost 1,200 people living with HIV (PLWH). Demographic, immunological and virological information on the study participants was compiled and antibody levels against SARS-CoV-2 spike (S) and receptor binding domain (RBD) were measured with commercial quantitative ELISA kits. Study subjects were grouped on the basis of absolute CD4+ T lymphocyte counts and their CD4+/CD8+ T cell ratios. Consistent with other studies, the authors report that lower CD4+ T cell counts are associated with a greater likelihood of not reaching what are considered to be protective levels of antibodies against SARS-CoV-2 S protein. A similar, but statistically weaker association was found with lower CD4+/CD8+ T cell ratios. The statistical analyses appear to have been done correctly and the large size of the cohort inspires confidence in the findings. A number of limitations of the study are brought up and considered by the authors in their discussion. While the reported findings are not completely novel, the power and rigor of the study are strong. I have a number of minor comments that could improve the manuscript if addressed.
- The groupings based on CD4+T cell counts are well matched for virological parameters, but the groups with lower counts tended to be older, infected with HIV for longer and more likely to suffer at least 1 co-morbidity. The potential impact on the authors’ findings and analyses should be discussed.
We thank you the reviewer for this comment. Indeed, we agreed that confounding is an important issue although all models are already adjusted for both age and the extent of comorbidities. Regarding the time since HIV infection, this could only be calculated for a very small subset of participants with known date of seroconversion. We calculated the time since HIV diagnosis, but we have chosen not to control for this variable because it is known to be an inaccurate estimate of the duration of infection and would potentially introduce measurement error bias. In addition, the effect of the duration of HIV infection should be captured by the nadir CD4 count. We have added a sentence to explain our reason for not including time since HIV diagnosis in the models in the Discussion section (lines 334-339).
- If there was no significant association between nadir CD4+T cell counts and vaccine responses, the authors should discuss what this means in terms of immune recovery for PLWH.
Our exposure of interest was the most recent CD4 count measured prior to the vaccination cycle. This is correlated to the nadir CD4 count which is also a potential predictor of the response to vaccination (although conflicting results have been published) [Kim HN et al. Int J STD AIDS 2009; El Chaer F et al. Am J Med 2019, Lapointe HR et al The J Infect Dis. 2023, Jedicke N et al HIV Med 2022]. Therefore, nadir CD4 count was included in the model to control for potential confounding and its association with the outcome was not investigated further. We have now added a comment in the Discussion to clarify that the association with nadir CD4 count was not in the scope of this analysis (lines 317-320).
- There was a weaker association between CD4+/CD8+T cell ratios and vaccine responses than with absolute CD4+ T cell counts. The authors should discuss this and clarify how the 2 measures (CD4+/CD8+ T cell ratios and absolute CD4+ T cell counts) can each have relevant, but different meanings, especially when CD8+ T cell counts are elevated well above normal.
We thank the reviewer for this comment. We have now expanded the Introduction section to better explain our reasons for evaluating and comparing the predictive ability of the two HIV markers (lines 72-89).
- Subjects were classified as having been infected with SARS-CoV-2 on the basis of anti-nucleocapsid protein antibodies. As there is considerable cross-reactivity with the nucleocapsids of common coronaviruses (especially OC43), did the authors test for selective reactivity with SARS-CoV-2 nucleocapsid?
Unfortunately, we did not test for cross-reactivity with other common coronavirus in this study. Nevertheless, two out of the 3 assays for SARS-CoV-2 anti-N titration used in the analysis are also kits approved and used for standard diagnostic procedures: the Roche Elecsys® Anti-SARS-CoV-2 anti-N and the Abbott Architect® Anti-SARS-CoV-2 anti-N IgG referring a clinical specificity of 99.8% and 99.6%, respectively. Moreover, the specificity of Abbott Architect® Anti-SARS-CoV-2 anti-N IgG assay against serum from other respiratory illnesses, including common coronaviruses, reached 100% in a study performed in 2020 (Meschi S et al. 2020). The third and last kit used for detection of anti-nucleocapsid of SARS-CoV-2 is a panel of Meso Scale Discovery (MSD) V-Plex, although this is used only for research purposes, is a validated multiplex assay for multiple detection of antibodies to antigens from SARS-CoV-2, including anti-N (see list of References below).
It needs to be highlighted that, the published literature shows that, with an estimated prevalence of infection around 10% - similar to that seen in our study- the positive predictive value of all the 3 assays is very high (≥97%). Thus, also considering that during the SARS-CoV-2 pandemic the prevalence of other coronavirus in the population was extremely low, we believe that the risk of cross-reactivity is negligible. Nevertheless, we have added a sentence in the Discussion to address this limitation (lines 357-361) and also the list of assays used for anti-N quantification in the methods (lines 145-150).
- Inés RM et al. Performance of Elecsys Anti-SARS CoV-2 (Roche) and VIDAS Anti-SARS CoV-2 (Biomérieux) for SARS-CoV-2 Nucleocapsid and Spike Protein Antibody Detection. EJIFCC. 2022 Aug 8;33(2):159-165
- Evaluation of the Abbott SARS-CoV-2 IgG for the detection of anti-SARSCoV-2 antibodies. Public Health England. Available at: https://assets.publishing.service.gov.uk/media/5eddf126e90e071b767bfce6/Evaluation_of_Abbott_SARS_CoV_2_IgG_PHE.pdf
- Turbett SE et al. Evaluation of Three Commercial SARS-CoV-2 Serologic Assays and Their Performance in Two-Test Algorithms. J Clin Microbiol. 2020 Dec 17;59(1):e01892-20
- Castro MDM et al. Performance verification of the Abbott SARS-CoV-2 test for qualitative detection of IgG in Cali, Colombia. PLoS One. 2021 Sep 1;16(9):e0256566
- Li FF et al A novel multiplex electrochemiluminescent immunoassay for detection and quantification of anti-SARS-CoV-2 IgG and anti-seasonal endemic human coronavirus IgG. J Clin Virol. 2022 Jan;146:105050
- Meschi S, et al. Performance evaluation of Abbott ARCHITECT SARS-CoV-2 IgG immunoassay in comparison with indirect immunofluorescence and virus microneutralization test. J Clin Virol. 2020 Aug;129:104539
- On line 78, it states a lower level of anti-S antibodies than anti-RBD antibodies was associated with protection (298 versus 775 BAU). This doesn’t make sense to me if the neutralizing antibodies are against the RBD. Please verify the accuracy of the numbers. On lines 82 and 83, a different study is cited with an opposite and more rational finding of a higher level of anti-S than anti-RBD antibodies being required for a protective effect (1591 versus 870 BAU).
We have checked the cited references and, although possibly counterintuitive, the thresholds used in the analysis are correct. It needs to be clear that the context in which these values were derived, and their significance is different. The first set of thresholds refer to vaccine efficacy in vivo. In contrast, the second specifically refer to average neutralizing response and were derived in laboratory. While vaccine efficacy certainly correlates with neutralizing response and neutralizing antibodies are mainly, but not exclusively, directed against the RBD domain of the S protein, additional immune markers contribute to VE. Indeed, markers of mucosal immunity, innate and adaptive cellular immune response notably participate in determining the efficacy of COVID-19 vaccines or in natural response against infection. Therefore, it is somewhat not so unexpected to have incongruent thresholds in the two studies. We have added two sentences in the Discussion and Conclusions regarding the differences in the thresholds (lines 377-380 and 388-391)
- A number of studies have suggested that the Moderna SARS-CoV-2 mRNA vaccine formulation may be slightly more immunogenic than the Pfizer SARS-CoV-2 mRNA vaccine formulation. Can the authors address this possibility with the data collected?
We thank the reviewer for this valuable suggestion. Much literature has been produced comparing the immunogenicity (and efficacy) of different vaccine platforms or vaccine types (mRNA vs. viral vector, mRNA-1273 vs. BNT162b2) and different vaccine sequencing strategies for boosting (heterologous vs homologous - see a list of selected References below). Indeed, the VAXICONA-ORCHESTRA study represents an ideal setting also to investigate these associations in the HIV-population. However, considering the aims of the current analysis (focussing on CD4 count and CD4/CD8 ratio as the main exposures) and the complications associated with properly evaluating this additional research hypothesis, we believe that it should be addressed in a separate full paper.
- Hermosilla E et al. Comparative effectiveness and safety of homologous two-dose ChAdOx1 versus heterologous vaccination with ChAdOx1 and BNT162b2. Nat Commun. 2022 Mar 23;13(1):1639
- Mues KE et al. Real-world comparative effectiveness of mRNA-1273 and BNT162b2 vaccines among immunocompromised adults identified in administrative claims data in the United States. Vaccine. 2022 Nov 8;40(47):6730-6739
- González de Aledo M et al. Safety and Immunogenicity of SARS-CoV-2 vaccines in people with HIV. AIDS. 2022 Apr 1;36(5):691-695
- Wang L, et al. Comparison of mRNA-1273 and BNT162b2 Vaccines on Breakthrough SARS-CoV-2 Infections, Hospitalizations, and Death During the Delta-Predominant Period. 2022;327(7):678–680]
- Ioannou GN et al. Comparison of Moderna versus Pfizer-BioNTech COVID-19 vaccine outcomes: A target trial emulation study in the U.S. Veterans Affairs healthcare system. EClinicalMedicine. 2022 Mar 5;45:101326
- Bánki Z, et al. Heterologous ChAdOx1/BNT162b2 vaccination induces stronger immune response than homologous ChAdOx1 vaccination: The pragmatic, multi-center, three-arm, partially randomized HEVACC trial. EBioMedicine. 2022 Jun;80:104073.

Reviewer 2 Report
Comments and Suggestions for Authors
- Major comments:
The research titled "SARS-CoV-2 mRNA vaccines response in people living with HIV according to CD4 count and CD4/CD8 ratio" offers valuable insights into the efficacy of COVID-19 mRNA vaccines among people living with HIV (PLWH). One of the study's strengths is its comprehensive sample size, encompassing 1,176 PLWH, providing a robust dataset for analysis. The research conclusively found that the CD4 count, more than the CD4/CD8 ratio, is a reliable predictor of the humoral response to the vaccine. Notably, a third vaccine dose significantly enhanced the vaccine response, especially in PLWH with higher CD4 metrics.
The study's conclusions underscore the importance of considering individual immunological markers, like CD4 counts, when assessing vaccine efficacy in PLWH. This research contributes significantly to the field by highlighting the potential need for tailored vaccination strategies for PLWH, emphasizing the pivotal role of the third vaccine dose. By bridging the knowledge gap on vaccine responses in PLWH, this study paves the way for more personalized medical interventions and can influence future vaccination guidelines for this specific population.
- General concept comments
Here are some considerations for the study:
- Observational Nature: The study is observational, which means there's a potential for unmeasured confounding bias that might not have been accounted for.
- Serology Assays Variation: Different serology assays were used across various centers. This introduces potential issues related to standardization and interpretability of results.
- Focus on Specific Variants: The study period mainly covered the alpha and delta variants of COVID-19. This suggests that the results might differ for other variants, such as the Omicron variant.
- Lack of Control Group: The study exclusively focused on people living with HIV (PLWH) and did not include a control group of HIV-negative individuals. This means the study cannot estimate the effect of HIV infection per se but only compares responses over time according to HIV-related immunological markers.
5. The study provides references and citations, but they do not offer a comprehensive view of the background information presented in the article's introduction. To provide enough background information, the following points should be considered, such as:
· The current state of knowledge about the topic.
· The significance of studying the humoral response to SARS-CoV-2 vaccines in people living with HIV.
· Previous research findings related to the topic.
· The rationale for the study's specific focus on CD4 count and CD4/CD8 ratio.
These limitations highlight areas where caution should be exercised when interpreting the study's results and suggest avenues for further research to address these gaps.
- Specific comments:
a. Line 40, what does “BAU” mean here?
b. Line 39-41, why did you choose the two criteria?
c. Line 69, typo of “CD/CD8 ratio.”
d. Line 77, what does “COVE” mean here?
e. Line 108-109, CD4 and CD8 cell count before 1st dose or within 1-month from 1st dose? Please list the rationale for including CD4 and CD8 cell counts within 1-month from 1st dose, which could differ from the samples before the 1st dose.
f. Line 153, what does “WP4” mean here?
g. What does “ART” mean in Table 1?
h. Line 202, typo of “31.9nd 37.8”.
Comments on the Quality of English LanguageExtensive editing of the English language is required.
Author Response
Please see the attachment.
- Major comments:
The research titled "SARS-CoV-2 mRNA vaccines response in people living with HIV according to CD4 count and CD4/CD8 ratio" offers valuable insights into the efficacy of COVID-19 mRNA vaccines among people living with HIV (PLWH). One of the study's strengths is its comprehensive sample size, encompassing 1,176 PLWH, providing a robust dataset for analysis. The research conclusively found that the CD4 count, more than the CD4/CD8 ratio, is a reliable predictor of the humoral response to the vaccine. Notably, a third vaccine dose significantly enhanced the vaccine response, especially in PLWH with higher CD4 metrics.
The study's conclusions underscore the importance of considering individual immunological markers, like CD4 counts, when assessing vaccine efficacy in PLWH. This research contributes significantly to the field by highlighting the potential need for tailored vaccination strategies for PLWH, emphasizing the pivotal role of the third vaccine dose. By bridging the knowledge gap on vaccine responses in PLWH, this study paves the way for more personalized medical interventions and can influence future vaccination guidelines for this specific population.
- General concept comments
Here are some considerations for the study:
- Observational Nature: The study is observational, which means there's a potential for unmeasured confounding bias that might not have been accounted for.
We had acknowledged unmeasured confounding as important limitation in the Discussion of the submitted version (lines 334-339)
- Serology Assays Variation: Different serology assays were used across various centers. This introduces potential issues related to standardization and interpretability of results.
We agree that the potential impact of using different assays on the results needed to be thoroughly investigated. Indeed, first of all, while we were preparing this revision, we realized that for some of the participants the labelling of the assay used has been inverted (the assay used was Roche but it was misclassified as DiaSorin). We have now corrected this issue and the new results are similar. In addition, as pointed out by another reviewer, Elecsys® anti-SARS-Cov-2 assay by Roche measures total Ig, not just IgG so its results are not directly comparable to those produced with DiaSorin. We have now performed additional sensitivity analyses after excluding samples which were tested with the Roche assays and, again, results were similar.
- Focus on Specific Variants: The study period mainly covered the alpha and delta variants of COVID-19. This suggests that the results might differ for other variants, such as the Omicron variant.
This aspect was also extensively discussed in the original submitted version.
- Lack of Control Group: The study exclusively focused on people living with HIV (PLWH) and did not include a control group of HIV-negative individuals. This means the study cannot estimate the effect of HIV infection per se but only compares responses over time according to HIV-related immunological markers.
The original submission already included a specific sentence on this issue using almost the exact wording of the text above.
- The study provides references and citations, but they do not offer a comprehensive view of the background information presented in the article's introduction. To provide enough background information, the following points should be considered, such as:
- The current state of knowledge about the topic.
. The significance of studying the humoral response to SARS-CoV-2 vaccines in
people living with HIV.
- Previous research findings related to the topic.
- The rationale for the study's specific focus on CD4 count and CD4/CD8 ratio.
We have now updated our Medline research and expanded the Introduction section including the evidence from the few new studies put our work into context. In response to another referee, we have also now better explained our reasons for evaluating and comparing the predictive ability of CD4 count an CD4/CD8 ration in the context of HIV infection and vaccine response (lines 72-89).
These limitations highlight areas where caution should be exercised when interpreting the study's results and suggest avenues for further research to address these gaps.
We thank this reviewer for summarizing our work and pointing out the main limitations of our analysis (which we believe have been already extensively addressed in the Discussion section of our original submission – see specific points above). The use of several different assays is the consequence of the large multicentre-multinational cohort and this per se can also be seen as a strength. The new sensitivity analyses suggest that our conclusions are robust against these limitations.
- Specific comments:
- Line 40, what does “BAU” mean here? à
We have now spelled out BAU (as binding antibody unit) the first time it appears in both the Abstract and Introduction section.
- Line 39-41, why did you choose the two criteria? à
We have now better clarified the rationale for choosing the two sets of response thresholds (both in the Abstract and in the main text)
- Line 69, typo of “CD/CD8 ratio.” à
We thank the referee for spotting this typo which has now been corrected.
- Line 77, what does “COVE” mean here? à
It is the name of the trial which was also reported in the citation.
- Line 108-109, CD4 and CD8 cell count before 1st dose or within 1-month from 1st dose? Please list the rationale for including CD4 and CD8 cell counts within 1-month from 1st dose, which could differ from the samples before the 1stdose.
The decision to include values of CD4 and CD8 count measured within 1 month after the date of the first dose was made a priori in order to maximise the study sample size. However, a posteriori, only 7 (0.6%) of the 1,176 PLWH included in our analysis had the markers measured after the first vaccine dose. Also, another recent analysis of this dataset showed no evidence for an association between SARS-CoV-2 mRNA vaccination and significant transient CD4 count and CD8 count changes comparing values pre- vs. post-dose, suggesting that the inclusion of post vaccination values is unlikely to have introduced a bias ( https://www.icar2023.it/public/repository/slide/2_4_7_0_4_slide.pdf )
- Line 153, what does “WP4” mean here?
WP4 stands for Work Package 4 of the original grant application. The meaning has been now specified.
- What does “ART” mean in Table 1?
ART is a common acronym used to indicate antiretroviral treatment. We have now spelled out the abbreviation both in Table 1 and in the main text.
- Line 202, typo of “31.9nd 37.8”.
We thank the referee for spotting this typo which has now been corrected.
This manuscript is well conducted with a good statistical analysis. However, the serologic response outcomes are lacking. You can add more information about the Ig and IgG outcomes to make this manuscript more informative and beneficial to the reader because the serologic response in PLWH is scant information.
We thank the reviewer for this comment. The serologic response was indeed our main focus here. Also following the suggestion of another reviewer, we have now performed sensitivity analyses after restricting the logistic and mixed-liner regression only to IgG outcome values rather than total Ig responses. Also as suggested by another reviewer we have also performed a more detailed descriptive statistic such as the scatterplot of the serologic responses against CD4 count at the various time points of the vaccination schedule (after T1) which are included below.
The analysis confirms a general association between the HIV markers and serological response, but because of the attrition bias, the steeper slopes observed at time points after T3 are difficult to interpret. Moreover, the different higher/lower limits of quantifications of the different assays used, clearly identifiable in the scatter plots below, further make these correlations difficult to interpret. In addition, these simpler analyses do not consider the correlation of values coming from the same individual at different time-points which is instead properly accounted for in the linear-mixed model in the manuscript. Considering these issues, although these data reveal a trend which goes in the same direction as that seen in the main analysis they do not seem to add substantial information and we therefore decided to not include them in the main text/supplemental material.

Reviewer 3 Report
Comments and Suggestions for Authors
This manuscript is well conducted with a good statistical analysis. However, the serologic response outcomes are lacking. You can add more information about the Ig and IgG outcomes to make this manuscript more informative and beneficial to the reader because the serologic response in PLWH is scant information.
Major concerns.
1. Please check your immunologic assessment tool used, Elecsys® Anti-SARS-CoV-2.
This reagent detects total Ig anti-nucleocapsid. The outcomes may be almost seronegative, except for convalescent or whole virion-inactivated vaccines (e.g. CoronaVac, BBIBP-CorV).
- All anti-SARS-CoV-2 reagents detect immunoglobulin (Total Ig), not only IgG.
- Elecsys® Anti-SARS-CoV-2 is targeting Ig anti-nucleocapsid. The unit is COI (semi-quantitative). https://diagnostics.roche.com/global/en/products/params/elecsys-anti-sars-cov-2.html
- Elecsys® Anti-SARS-CoV-2 S is targeting Ig anti-RBD. The unit is U/mL (quantitative). https://diagnostics.roche.com/global/en/products/params/elecsys-anti-sars-cov-2-s.html
2. The Endpoint, VE. The VE criteria used in this manuscript are based on the ancestral, A.1 (clade S) or related strains.
Nowadays, the prominent circulation is Omicron. The vaccine efficacy/effectiveness is substantially reduced.
Suggest adding the statement of the VE used to clarify the protection and add to the discussion.
Minor concerns.
1. Suggest adding the conversion factor to calculate the BAU/mL because some publications or manuals may have slightly different conversion factors. For example, Elecsys® Anti-SARS-CoV-2 S may be used 1.03 or 1.028.
Comments.
1. You can create a table to summarise each serologic assessment used in this study by; total and stratified by your criteria (LCD4, ICD4, HCD4, LR, IR, HR) to give your paper more information. This table may be added to the supplementary materials.
2. Have you tried to create a correlation between each serologic and CD4 level?
The XY plot may reveal the trend of the B cell response (Ig or IgG) and CD count or ratio.
Author Response
Please see the attachment
Major concerns.
- Please check your immunologic assessment tool used, Elecsys® Anti-SARS-CoV-2.
This reagent detects total Ig anti-nucleocapsid. The outcomes may be almost seronegative, except for convalescent or whole virion-inactivated vaccines (e.g. CoronaVac, BBIBP-CorV).
Indeed, Elecsys® anti-SARS-Cov-2 by Roche if available for targeting both anti- nucleocapsid and anti-RBD of spike, the first one has been used only to establish whether the participants had been previously infected with SARS-CoV2 while the second to estimate the humoral response.
- All anti-SARS-CoV-2 reagents detect immunoglobulin (Total Ig), not only IgG.
- Elecsys® Anti-SARS-CoV-2 is targeting Ig anti-nucleocapsid. The unit is COI (semi-quantitative). https://diagnostics.roche.com/global/en/products/params/elecsys-anti-sars-cov-2.html
- Elecsys® Anti-SARS-CoV-2 S is targeting Ig anti-RBD. The unit is U/mL (quantitative). https://diagnostics.roche.com/global/en/products/params/elecsys-anti-sars-cov-2-s.html
The comment of the reviewer is very relevant as some of our original analysis indeed mixed results of the Roche, MSD and Abbott assays (which measures total Ig) with those measured using DiaSorin (which only measures IgG).
We therefore decided to add a series of sensitivity analyses in which values measured with the Elecsys® Anti-SARS CoV-2 S were excluded. The results of these additional analyses were similar to those of the main analysis. Of note, while preparing this revision we also realized that for some of the participants the labelling of the assay used has been inverted (the assay used was Roche but it was misclassified as DiaSorin) and this explains why also the results of the analysis including the values measured with both Roche and DiaSorin are slightly different from those originally submitted. Reassuringly, results were similar after this correction. Results and Discussion sections have been modified accordingly.
Regarding the use of anti-nucleocapsid detection to exclude participants who had previously been infected with SARS Cov2, the measurements of the Elecsys® assays have been used only as qualitative results (binary response: Previously infected/Not previously infected with SARS-CoV-2), and therefore the issue of different targeting (IgG vs. total Ig) does not constitute a problem for this part of our analysis.
- The Endpoint, VE. The VE criteria used in this manuscript are based on the ancestral, A.1 (clade S) or related strains. Nowadays, the prominent circulation is Omicron. The vaccine efficacy/effectiveness is substantially reduced. Suggest adding the statement of the VE used to clarify the protection and add to the discussion.
We agree with the reviewer, and we have added this sentence in the limitations (lines 350-353).
Minor concerns.
- Suggest adding the conversion factor to calculate the BAU/mL because some publications or manuals may have slightly different conversion factors. For example, Elecsys® Anti-SARS-CoV-2 S may be used 1.03 or 1.028.
We thank the reviewer for this relevant suggestion. We have now added in the Supplementary materials, the conversion rates used for all the assays.
Comments.
- You can create a table to summarise each serologic assessment used in this study by; total and stratified by your criteria (LCD4, ICD4, HCD4, LR, IR, HR) to give your paper more information. This table may be added to the supplementary materials.
We thank the reviewer for this comment. We have now summarized in the new Supplementary Figure 1, median and IQR of anti-S/anti-RBD titres at various time points of the vaccination schedule (after T1) and stratified according to the immunoassay used.
Supplementary Table 5 shows the same values, further stratified also by the exposure categories for CD4 count and CD4/CD8 ratio of Roche and DiaSorin assays (which are those included in the analysis with a continuous outcome) are also included in the main text.
- Have you tried to create a correlation between each serologic and CD4 level?
The XY plot may reveal the trend of the B cell response (Ig or IgG) and CD4 count or ratio.
Please find shown below the suggested scatterplots using the data at various points of the vaccination schedule after T1 (first dose). The analysis confirms a general association between the HIV markers and serological response, but because of the attrition bias, the steeper slopes observed at time points after T3 are difficult to interpret. Moreover, the different higher/lower limits of quantifications of the different assays used, clearly identifiable in the scatter plots below, further make these correlations difficult to interpret. In addition, these simpler analyses do not consider the correlation of values coming from the same individual at different time-points which is instead properly accounted for in the linear-mixed model in the manuscript. Considering these issues, although these data reveal a trend which goes in the same direction as that seen in the main analysis they do not seem to add substantial information and we therefore decided to not include them in the main text/supplemental material.

Round 2
Reviewer 2 Report
Comments and Suggestions for Authors
The authors of the study have diligently addressed my concerns. They clarified methodological choices, acknowledged the study's observational nature, and rectified typographical errors. The introduction was expanded for greater context, and emphasis was placed on serologic response outcomes with additional analyses. Their thorough responses, combined with the recognition of the manuscript's robust statistical analysis, indicate a commitment to quality and suggest the study's potential value to the field. I would consider recommending the manuscript for publication.
Comments on the Quality of English LanguageFine.